# Comparison of Two Methods for Measuring the Temperature Dependence of H_2_ Permeation Parameters in Nitrile Butadiene Rubber Polymer Composites Blended with Fillers: The Volumetric Analysis Method and the Differential Pressure Method

**DOI:** 10.3390/polym16020280

**Published:** 2024-01-19

**Authors:** Ji Hun Lee, Ye Won Kim, Do Jung Kim, Nak Kwan Chung, Jae Kap Jung

**Affiliations:** 1Hydrogen Energy Materials Research Team, Korea Research Institute of Standards and Science, Daejeon 34113, Republic of Korea; ljh93@kriss.re.kr (J.H.L.); kyw9687@kriss.re.kr (Y.W.K.); dojung@kriss.re.kr (D.J.K.); 2Department of Measurement Science, University of Science and Technology, 217 Gajeong-ro, Yuseong-gu, Daejeon 34113, Republic of Korea; 3Department of Material Science and Engineering, Chungnam National University, 99, Daehak-ro, Yuseong-gu, Daejeon 34134, Republic of Korea

**Keywords:** activation energy, hydrogen diffusion, NBR polymer composites, temperature dependence, volumetric analysis method, differential pressure method

## Abstract

Hydrogen uptake/diffusivity in nitrile butadiene rubber (NBR) blended with carbon black (CB) and silica fillers was measured with a volumetric analysis method in the 258–323 K temperature range. The temperature-dependent H_2_ diffusivity was obtained by assuming constant solubility with temperature variations. The logarithmic diffusivity decreased linearly with increasing reciprocal temperature. The diffusion activation energies were calculated with the Arrhenius equation. The activation energies for NBR blended with high-abrasion furnace CB and silica fillers increased linearly with increasing filler content. For NBR blended with medium thermal CB filler, the activation energy decreased with increasing filler content. The activation energy filler dependency is similar to the glass transition temperature filler dependency, as determined with dynamic mechanical analysis. Additionally, the activation energy was compared with that obtained by the differential pressure method through permeability temperature dependence. The same activation energy between diffusion and permeation in the range of 33–39 kJ/mol was obtained, supporting the temperature-independent H_2_ solubility and H_2_ physisorption in polymer composites.

## 1. Introduction

The diffusion coefficient in a polymer composite is a function of the shape and size of the penetrant molecule, the kinetic diameter of the gas molecule, and the filler species [1,2]. The diffusion characteristics of gas molecules through polymers are very important for various scientific and engineering fields, such as the medical, textile, membrane separation, food packaging, solvent extraction, contaminant extraction, and gas sealing industries [3,4,5,6].

Furthermore, diffusion and permeation of rubbery polymers are the main controlling parameters for low-permeation applications such as O-ring seals under high hydrogen pressures. Thus, many studies have been conducted on diffusion/permeation improvements with various filler species, filler contents, crosslinking agents, and additives in O-ring candidate materials, such as nitrile butadiene rubber (NBR), ethylene propylene diene monomer (EPDM), and fluoroelastomer (FKM) [7,8,9,10,11,12,13,14,15]. However, experimental investigations of the temperature dependence of H_2_ diffusion and permeation are comparatively rare. Studies on the temperature dependence of permeation parameters are essential for clarifying the mechanisms and related dynamics of polymeric materials. In particular, the dynamic properties of hydrogen transport were clarified in harsh environments with wide temperature changes (−50 °C to 90 °C) for actual use in the hydrogen infrastructure. The temperature dependence of gas transport through a polymer membrane is normally described by the Arrhenius equation.

Thus, we have recently established two temperature-dependent measuring systems for H_2_ diffusion/permeation together with uncertainty analysis. The H_2_ diffusion coefficients and absorbed contents of the investigated composites were determined in the temperature range above the glass transition temperature with a volumetric analysis method (VAM). This is a method for measuring the released H_2_ concentration and diffusivity after a specimen exposed to high-pressure hydrogen is decompressed to atmospheric pressure. This is based on the principle of volume measurement, where H_2_ released from the specimen causes a decrease in the water level in the graduated cylinder with increasing elapsed time. The diffusion coefficients were obtained from the kinetics for absorption/desorption [16] into the gas phase with the VAM and a modified diffusion analysis program. The activation energy and the pre-exponential factor for diffusion were determined by the temperature dependence of the diffusivity. The effects of fillers on the activation energies of NBR composites were analyzed by considering the roles of fillers. In addition, the activation energies were compared with those obtained by permeation measurements with the differential pressure method (DPM). This method measures the differential pressure between two vertically placed cells, separated by testing sheet specimens. The upside cell is the high feed concentration side, which receives the testing gas from the gas tank. The downside cell is the lower permeate concentration side, which receives the permeating gas through specimens and measures the pressure of the gas with a pressure detector. Thus, the increase in pressure in the downside cell is converted into an increased H_2_ concentration, resulting in the diffusivity/permeation measurement. The temperature-independent solubility levels obtained from the two methods were described by considering the activation energy and physisorption characteristics of H_2_. The activation energy dependency on the filler was similar to the glass transition temperature dependency on the filler, as determined with a dynamic mechanical analysis (DMA).

## 2. Experiments and Analyses

We first describe the measurements and analysis principles for the temperature-controlled system based on the VAM in Section 2.1, Section 2.2, Section 2.3 and Section 2.4. The temperature-controlled system based on the measurement principle of the DPM is then briefly described in Section 2.5.

### 2.1. High-Pressure Hydrogen Exposure of the Specimens in the Volumetric Analysis Method

To determine the temperature dependence of the diffusion coefficient, we used polymer specimens, such as the NBR polymer composites used in O-ring seals. The compositions and production methods for polymer specimens blended with high-abrasion furnace (HAF) carbon black (CB), medium thermal (MT) CB, and silica fillers were reported in the literature [17]. Before high-pressure H_2_ exposure, the NBR composites were degassed by heating at 60 °C for 48 h in the temperature chamber.

To conduct high-pressure H_2_ exposure and subsequent decompression in the high-pressure chamber, a SUS 316 chamber (Figure 1a) with an inner diameter of 70 mm and a height of 120 mm was used at room temperature and at a specified pressure [18]. The chamber was purged three times with H_2_ gas at 1 MPa before H_2_ exposure. We exposed the specimen to hydrogen at a high pressure (7 MPa) for 24 h at room temperature. Hydrogen gas charging for 24 h was sufficient to reach gas sorption equilibrium for the diffusion measurements. After exposure to H_2_, the valve was opened, and the H_2_ in the chamber was released. After decompression, the elapsed time was recorded from the moment (*t* = 0) at which the high-pressure H_2_ in the chamber was reduced to atmospheric pressure.

### 2.2. H_2_ Emissions Measured by the Volumetric Analysis Method

Figure 1 illustrates the VAM for measuring the concentration of the emitted H_2_ at room temperature, which consists of a high-pressure chamber for H_2_ exposure (Figure 1a) and a graduated cylinder partially immersed in a water container (Figure 1b) [7].

After exposure to the high-pressure chamber and subsequent decompression, the specimens were loaded into the loading cell connected to the graduated cylinder with a stainless steel (SUS) tube, as shown in Figure 1b. After decompression, the H_2_ released from the specimens gradually reduced the water level (*h*) in the graduated cylinder over time. Therefore, the pressure (P) and volume (V) of the hydrogen gas inside the graduated cylinder changed with time.

The hydrogen gas inside the cylinder was governed by the ideal gas equation, *PV* = *nRT*, where R is the ideal gas constant (8.20544 × 10^−5^ m^3^·atm/(mol·K)), *T* is the temperature of the gas occupied in the top of the graduated cylinder, and *n* is the number of H_2_ moles released into the cylinder. The time-dependent pressure P(t) and volume $V\left(t\right)$ of the gas inside the graduated cylinder were expressed as follows:(1)$$P(t)=P_{o}-\rho gh(t),V(t)=V_{o}-V_{s}-V_{h}(t)$$
where $P_{o}$ is the atmospheric pressure outside the cylinder, ρ is the density of the distilled water in the water container, *g* is the gravitational acceleration, h(t) is the level (or height) of the water volume inside the cylinder measured from the water level in the water container, $V_{o}$ is the total combined volume of gas and water inside the graduated cylinder measured from the water level in the water container, $V_{h}(t)$ is the time-dependent water volume inside the graduated cylinder measured from the water level in the water container, and $V_{s}$ is the volume of the specimen.

The H_2_ content released from the specimen was determined by the water volume $\left[V_{h}\left(t\right)\right]$ over time. Thus, the total moles [n(t)] of released H_2_ were determined by measuring the total gas volume [V(t)] in the graduated cylinder as the reduction in the water level [7,18].
$$n\left(t\right)=\frac{P\left(t\right)V\left(t\right)}{RT\left(t\right)}=\frac{P\left(t\right)\left[V_{a}+V_{H}\left(t\right)\right]}{RT\left(t\right)}=\frac{P_{0}[1+\beta \left(t\right)][V_{a}+V_{H}\left(t\right)]}{RT_{0}[1+\alpha \left(t\right)]}$$
(2)$$≅\frac{P_{0}}{RT_{0}}\left[V_{a}+V_{H}\left(t\right)+V\left(t\right)\left(\beta (t)-\alpha (t)\right)\right]=n_{a}\left(t\right)+n_{H}\left(t\right),$$
$$n_{a}\left(t\right)=\frac{P_{0}}{RT_{0}}V_{a},n_{H}\left(t\right)=\frac{P_{0}}{RT_{0}}\left[V_{H}(t)+V\left(t\right)\left(\beta (t)-\alpha (t)\right)\right]$$
$$\alpha \left(t\right)=\frac{T(t)-T_{0}}{T_{0}},\;\beta \left(t\right)=\frac{P(t)-P_{0}}{P_{0}}.$$
where $T_{0\;}$ and $P_{0}$ are the initial temperature and pressure of the gas inside the cylinder, respectively, V(t) is the sum of the initial remaining air volume ($V_{a})$ and the emitted hydrogen volume ${[V}_{H}\left(t\right)]$, i.e., $V\left(t\right)=V_{a}+V_{H}\left(t\right)$, $n_{a}$ is the initial number of moles of air, and $n_{H}(t)$ is the moles of H_2_ corresponding to the hydrogen volume emitted at time *t*. Thus, $n_{H}(t)$ was converted into the H_2_ concentration emitted [$C\left(t\right)]$ per mass from the rubber:$$C\left(t\right)\left[wt\cdot ppm\right]=n_{H}\left(t\right)\left[mol\right]\times \frac{m_{H2}\left[\frac{g}{mol}\right]}{m_{specimen}\left[g\right]}\times 10^{6}$$
(3)$$=\frac{P_{0}}{RT_{0}}\left[V_{H}(t)+V\left(t\right)\left(\beta (t)-\alpha (t)\right)\right]\left[mol\right]\times \frac{m_{H2}\left[\frac{g}{mol}\right]}{m_{specimen}\left[g\right]}\times 10^{6}$$
where $\;m_{H2}$ [g/mol] is the molar mass of H_2_ ($m_{H2}$ [g/mol] = 2.016 g/mol), $m_{specimen}$ is the mass of the specimen, and $n_{H}(t)$ and $C\left(t\right)$ are factors directly affected by variations in the temperature and pressure. For precise measurements, we compensated for the changes in temperature and pressure.

### 2.3. Temperature-Controlled System in the Volumetric Analysis Method

We established a system for measuring the temperature dependence of H_2_ diffusivity in NBR polymer composites by employing the VAM. The system shown in Figure 2 simultaneously measured six specimens; the system consisted of a proportional integral differential (PID) temperature control chamber (TC) for measurements at the desired temperature, six specimens with cells inside the TC, six parallel graduated cylinders outside the TC, and a temperature–pressure meter/timer outside the TC. The hydrogen uptake and diffusivity in the system could be measured at temperatures ranging from 233 K to 363 K, with a stability of 0.2 K. The differences in the temperatures inside and outside the temperature control chamber were solved by compensating for the content emitted from the specimen; that is, they were solved by subtracting the water level (or gas volume) of an empty cylinder connected to a cell without a specimen from that of a similar system with a specimen. For the gas flow transfer emitted from the specimen, a SUS tube with an outer diameter of 1/8″ was connected with a faced rubber seal between the graduated cylinder and the specimen-containing cell.

### 2.4. Analysis of the Hydrogen Uptake and Diffusivity for the Volumetric Analysis Method

By assuming that H_2_ emission was a Fickian diffusion process, the mass concentration $C_{E}(t)$ of the emitted H_2_ was computed as follows [19,20]:$$C_{E}(t)/C_{\infty }=1-\frac{32}{\pi^{2}}\times \left[\sum_{n=0}^{\infty }\frac{exp\left\{\frac{{-\left(2n+1\right)}^{2}\pi^{2}Dt}{l^{2}}\right\}}{{\left(2n+1\right)}^{2}}\right]\times \left[\sum_{n=1}^{\infty }\frac{exp\left\{-\frac{D\beta_{n}^{2}t}{\rho^{2}}\right\}}{\beta_{n}^{2}}\right]$$
$$=1-\frac{32}{\pi^{2}}\times \left[\frac{\exp\left(-\frac{\pi^{2}Dt}{l^{2}}\right)}{1^{2}}+\frac{\exp\left(-\frac{3^{2}\pi^{2}Dt}{l^{2}}\right)}{3^{2}}+\ldots ,+\frac{\exp\left(-\frac{{\left(2n+1\right)}^{2}\pi^{2}Dt}{l^{2}}\right)}{(2{n+1)}^{2}}+\ldots ,\right]$$
(4)$$\times \left[\frac{\exp\left(-\frac{D\beta_{1}^{2}t}{\rho^{2}}\right)}{\beta_{1}^{2}}+\frac{\exp\left(-\frac{D\beta_{2}^{2}t}{\rho^{2}}\right)}{\beta_{2}^{2}}+\ldots ,+\frac{\exp\left(-\frac{D\beta_{n}^{2}t}{\rho^{2}}\right)}{\beta_{n}^{2}}+\ldots ,\right]$$
where $\beta_{n}$ is the root of the zeroth-order Bessel function J_0_(β_n_) with β_1_ = 2.40483, β_2_ = 5.52008, β_3_ = 8.65373, …, β_50_ = 156.295. Equation (4) is an infinite series expansion with two summations. The equation provides the solution for Fick’s second diffusion equation for a cylindrical polymer specimen. Herein, C_E_ = 0 at *t* = 0 and C_E_ =$C_{\infty }$ at *t* = ∞. $C_{\infty }$ is the saturated H_2_ concentration at infinite time, i.e., the H_2_ uptake, D is the diffusion coefficient, and l and ρ are the thickness and radius of the cylindrical specimen, respectively.

Because Equation (4) has a complicated form with two infinite summation terms, a dedicated diffusion analysis program was needed to calculate D and $C_{\infty }$. By applying the diffusion analysis program developed from the Nelder–Mead simplex nonlinear optimization algorithm [21,22], we analyzed the time-dependent emission data, $C_{E}\left(t\right)$, with Equation (4). Therefore, D and $C_{\infty }$ were obtained. The program and its detailed procedure were described in the literature [18].

### 2.5. Permeation Cell and Temperature-Controlled System in the Differential Pressure Method

The DPM followed Fick’s diffusion law and Henry’s gas solubility law, while the permeability (*P*), *D*, and solubility (*S*) could be calculated with these laws [11]. Figure 3a,b show the permeation cell with a mounted sample and the schematics of the overall temperature-controlled DPM system, respectively, based on ISO 15105-1 [23].

A disk-shaped sample with a diameter of 35 mm was placed on the lower sample holder, as shown in Figure 3a. O-rings, bolts, and vacuum grease were used to ensure tight sealing between the sample and the two holders. The inside of the assembled permeation cell was evacuated for approximately 2 days with a vacuum pump until the outgassing rate was reduced to less than 1 × 10^−7^ Pa/s. After outgassing, we closed the two valves connected to the upper and lower pumps. When the pressure increase rate reached a constant steady state for the elapsed time, the valve connected to the high-pressure H_2_ storage vessel was opened to fill the hydrogen gas to a pressure of 0.1 MPa, and the gradual increase in the pressure on the permeate side of the cell (lower cell) was measured. The constant pressure increase rate before hydrogen injection was subtracted from the measured permeation curve to determine the effect of hydrogen permeation.

As shown in Figure 3b, with the bath fluid circulating inside the permeation cell and thermal insulating jacket, the temperature setting for the circulator ranged from 203 K to 423 K with a good stability of 0.1 K during the measurement. The bath fluid was circulated to maintain the desired temperature inside the cell. The bath fluids were either ethanol or dimethyl silicone oil, depending on the experimental temperature range. Ethanol was used for experimental temperatures ranging from 203 K to 273 K, while dimethyl silicone oil was used for experimental temperatures ranging from 273 K to 423 K. A k-type thermocouple was inserted into the circulating position of the bath fluid to measure the temperature of the sample inside the sample holders. A thermal insulating jacket composed of glass fibers covered the entire permeation cell to minimize heat loss, as shown in Figure 3b.

Moreover, the *dp*/*dt* for the permeation curve slowly changed, and the measurement was continued until the gas permeation curve reached a constant rate (steady state condition). From the permeation curve, the slope (*dp*/*dt*) was obtained, and then the permeability (*P*) was calculated as follows:(5)$$P=\frac{1}{RTP_{H2}}\times V_{c}\times \frac{dp}{dt}\times \frac{d}{A}$$
where *R* is the ideal gas constant, *T* is the temperature in the permeation cell [K], *P_H_*_2_ is the hydrogen pressure of the feed side in the upper cell [Pa], *V_c_* is the gas volume of the permeated side of the cell [m^3^], *d* is the thickness of a sample [m], and *A* is the H_2_ contact permeation area of the sample [m^2^].

*D* was deduced from the time lag value (θ) intersecting the x-axis (time axis) as follows:(6)$$D=\frac{d^{2}}{6\theta }$$

### 2.6. Dynamic Mechanical Analysis

The glass transition temperature (T_g_) was analyzed using a dynamic mechanical analysis (DMA, DMA 242E, NETZSCH, Selb, Germany), and the test mode is tension mode. The sample dimension is 10 mm (length), 5 mm (width), and 2 mm (thickness). The dynamic analysis was performed from −100 °C to 100 °C at heating rates of 3 °C/min, 5 °C/min, and 10 °C/min, with a fixed frequency of 1 Hz and an amplitude of 20 μm.

The process for obtaining T_g_ is as follows: firstly, the temperature of the inflection point can be obtained at each heating rate; secondly, it can be expressed as a straight line with a positive slope as a function of heating rate. Finally, the temperature corresponding to the extrapolated value of 0 °C/min was determined as the T_g_.

## 3. Results and Discussion

We first describe the results for the temperature dependence and activation energy obtained from the temperature dependence of H_2_ diffusivity by the VAM and glass transition temperature measured by the DMA in Section 3.1. Then, a comparison of the activation energies in NBR composites from the VAM and DPM is presented in Section 3.2.

### 3.1. Temperature Dependence and Activation Energy of H_2_ Diffusivity in the Volumetric Analysis Method

We used the temperature dependence measurement system in Figure 2 and measured the H_2_ emission versus time at six different temperatures for ten NBR polymer composites blended with CB and silica fillers. Figure 4 shows plots of the H_2_ emission content versus elapsed time for neat NBR at different temperatures. Since the temperature-dependent H_2_ emissions versus time plots for the nine filled NBR polymer composites revealed behaviors similar to those of neat NBR, we showed a representative result for the neat NBR in Figure 4. By applying the diffusion analysis program to Equation (4), the H_2_ diffusivities for neat NBR were obtained at six different temperatures by assuming the same hydrogen uptake (C_∞_ = 117.6 $\mu g_{H_{2}}/g_{specimen}$ at infinite time) at 298.2 K, as shown with the blue arrow in Figure 4.

The obtained diffusivities increased with increasing temperatures. Figure 5 shows plots of the diffusion coefficients as a function of reciprocal temperature for nine filled NBR polymer composites and one neat NBR sample. As shown in Figure 5, the diffusivity result for the neat NBR is included in all panels for comparison with those of the NBR composites with fillers. For the case of NBR filled with HAF CB, which is shown in Figure 5a, the diffusivity in the filled NBR composites gradually decreased with increasing filler content, implying that the HAF filler suppressed H_2_ diffusion. However, for the case of NBR composites filled with MT CB and silica in Figure 5b and Figure 5c, respectively, the presence and content of the filler did not cause appreciable changes in the diffusivity within the uncertainty estimated later. In other words, the diffusivities for MT and the silica-filled NBR composites were almost the same as for neat NBR.

The temperature dependence for gas transport through a polymer can be described by the Arrhenius equation [24,25,26]:*D* = *D*_0_ *exp* (−*E*_*d*_/*RT*)(7)
*S* = *S*_0_ *exp* (−*H*_*s*_/*RT*)(8)
*P* = *P exp* (−*E*_*p*_/*RT*) = *SD*(9)
where *D*_0_, *S*_0_, and *P*_0_ are the temperature-independent preexponential factors for diffusivity, solubility, and permeability, respectively; *R* is the universal gas constant at 8.314 (J/mol-K); *T* is the absolute temperature; *E_d_* and *E_p_* are the activation energies for diffusion and permeation, respectively; and *H_s_* is the heat of sorption. According to Equation (9), when the hydrogen solubility *S* is constant regardless of the temperature, *E_d_* and *E_p_* are equal.

As shown in Figure 5, the logarithmic diffusivities decreased linearly with increasing reciprocal temperature. The activation energies for diffusion calculated from the diffusivity data and Equation (7) are presented in Figure 6. For the NBR composites filled with HAF CB, the activation energy increased linearly with increasing filler content, implying that the filler suppressed H_2_ diffusion, as indicated in Figure 5a. In contrast, the activation energies for NBR composites blended with MT CB and silica appeared to change linearly with increasing filler contents, but they were similar to the activation energy for neat NBR within the uncertainty level. The increasing/decreasing rates for the activation energy per filler content in the filled NBR composites were expressed as follows:NBR with HAF CB series: slope = (0.13 ± 0.01) kJ/(mol·phr)NBR with MT CB series: slope = (−0.04 ± 0.02) kJ/(mol·phr)NBR with silica series: slope = (0.09 ± 0.04) kJ/(mol·phr)

**Figure 6 polymers-16-00280-f006:**
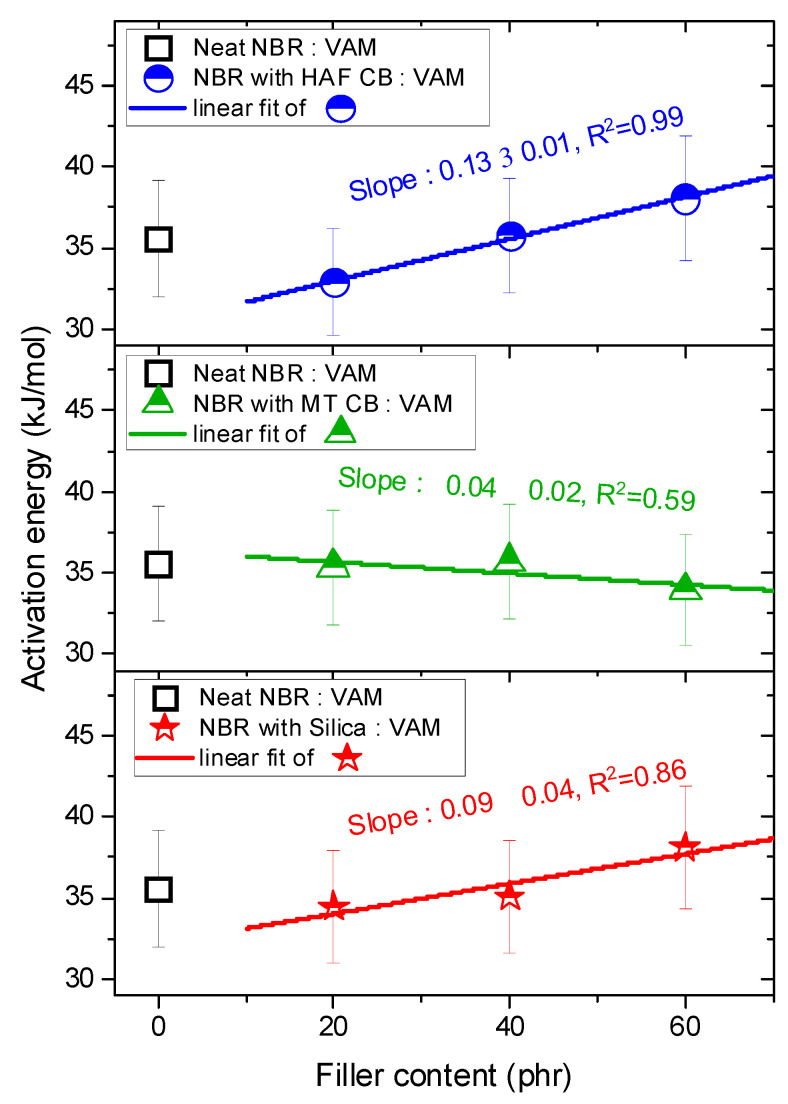
Activation energies versus filler contents for NBR composites blended with HAF, MT, and silica fillers. The legends show the linear least-squares fits of the activation energy versus filler content plots and their squared correlation coefficients, R^2^. The blue, green, and red lines indicate linear fits and the slopes for the NBR HAF CB series, the NBR MT CB series, and the NBR silica series, respectively. The error bars represent the expanded uncertainty (10%) of the activation energy, which will be estimated in the uncertainty analysis.

In the neat NBR, the H_2_ diffusion mechanism was only affected by the rubber chains (associated with the bulk diffusion mechanism). On the other hand, in a rubber-filler composite system, since the rubber chains are strongly physically adsorbed to the surface of the filler, the H_2_ diffusion mechanism is affected by the rubber chains and by the weakly connected pore-like rubber-filler interfacial chains (associated with the Knudsen diffusion mechanism) [27,28,29]. Thus, the mechanisms for H_2_ diffusion in the neat NBR were different from those of the filled NBR composites. Consequently, the activation energy for neat NBR was not included in the linear fits of the filled-NBR composites when investigating the filler effect.

Moreover, we measured the glass transition temperatures (T_g_) versus filler contents for the same specimens by employing the DMA. Figure 7 represents the filler-dependent T_g_ behavior, which was very similar to the filler-dependent E_d_ trend. The increasing or decreasing rates for the glass transitions per filler content were obtained from the following linear fits:NBR with HAF CB series: slope = (0.18 ± 0.05) °C/phrNBR with MT CB series: slope = (−0.07 ± 0.03) °C/phrNBR with silica series: slope = (0.10 ± 0.01) °C/phr

**Figure 7 polymers-16-00280-f007:**
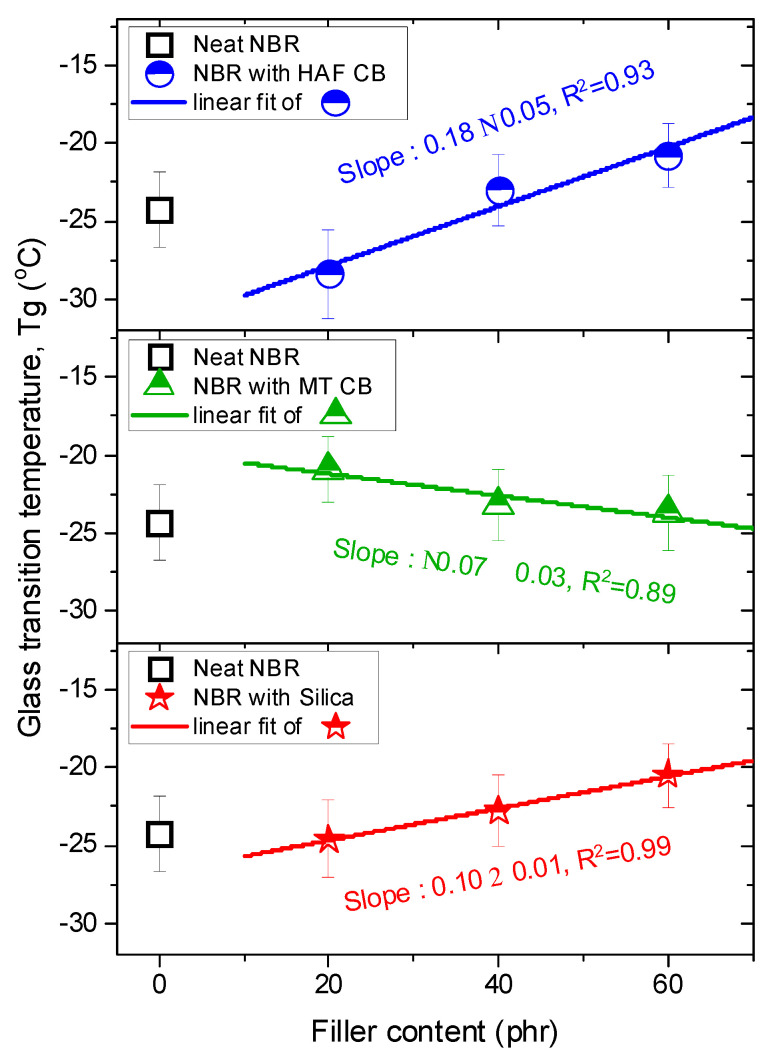
Plots of the glass transition temperature versus filler content for NBR composites blended with HAF CB, MT CB, and silica filler. The legends show the linear least-squares fits for the plots of T_g_ versus filler contents and their squared correlation coefficients, R^2^. The blue, green, and red lines indicate linear fits with slopes for the NBR with HAF CB series, the NBR with MT CB series, and the NBR with silica series, respectively.

The T_g_ was associated with the free volume, which was exponentially dependent on the diffusivity. Further investigations on the correlations and originality of these two similar behaviors for fractional free volume and diffusivity are needed.

### 3.2. Comparison of the Activation Energies from the Volumetric Analysis Method and the Differential Pressure Method

Figure 8 shows the hydrogen permeation curves for neat NBR measured with the DPM at five different temperatures. Since similar temperature-dependent results were observed for the nine filled NBR composites, we also displayed the representative hydrogen permeation curve for neat NBR in Figure 8. According to Equation (5), the H_2_ permeability of neat NBR was obtained from the corresponding slopes (*dp*/*dt*) at five different temperatures. Thus, the activation energy (*E_p_*) for permeation was determined according to Equation (9) from the slope of the permeation versus reciprocal temperature plot. Moreover, the H_2_ diffusivities of the neat NBR were calculated with Equation (6) at five different temperatures for the corresponding time lags. The activation energy for diffusion was obtained with Equation (7) from the slope of the diffusivity versus the reciprocal temperature plot.

To compare the activation energy obtained with the VAM with that obtained with the DPM, we selected a more precise measurement between the permeability and diffusivity. The permeability measurement in the DPM is more precise than the diffusivity measurement due to the comparatively large uncertainty in the time lag value (θ). Thus, we obtained the activation energy from permeability measurements in the DPM. In contrast, the diffusivity measurement in the VAM was more precise than the permeability measurement. Thus, we obtained the activation energy from diffusivity with the VAM.

The permeability measurement is different from the diffusivity measurement because the permeability contains the solubility and the diffusivity does not. However, under the assumption that the heat of sorption (*Hs*) in Equation (8) is zero, we concluded that the activation energies for permeability and diffusivity were the same. That is, when the hydrogen solubility S was constant regardless of the temperature, *E_d_* and *E_p_* were equal to each other according to Equation (9). As shown in Figure 9, we obtained the temperature-independent solubility with the DPM at two different pressures (0.1 MPa and 7 MPa) for neat NBR and NBR filled with HAF CB, MT CB, and silica at 40 phr. Figure 9a represents the solubilities measured at 0.1 MPa in the DPM for three different temperatures, indicating that the solubility was constant regardless of the temperature. We also measured the solubility at 7 MPa with the DPM for three different temperatures, as shown in Figure 9b, and concluded that the temperature-independent solubility was not affected by the pressure. Consequently, we can compare the activation energy for the diffusivity obtained with the VAM at 7 MPa with that for the permeability obtained with the DPM at 0.1 MPa.

The results for the two methods in ten NBR composites were consistent with each other within the expanded uncertainty level, as shown in Figure 10a–c. According to Equations (7)–(9), the same activation energies for diffusion and permeation suggested that the solubility was unchanged with temperature variations above T_g_; that is, our assumption was reasonable for obtaining the diffusivities at different temperatures with the VAM. The temperature-independent solubility was explained in the following manner: The diffused H_2_ did not undergo a chemical reaction with the polymer network as the temperature was varied, and it did not cause a structural change in the parent polymer. Additionally, the activation energies determined within the range of 33–39 kJ/mol implied that the adsorption of hydrogen was governed by physisorption rather than chemical sorption. These results were satisfactorily consistent with those in the prior literature [11,13,30].

## 4. Uncertainty Analyses

We first describe the uncertainty analysis for diffusivity with the VAM in Section 4.1 and then describe that of permeability with the DPM in Section 4.2. The uncertainty of the temperature dependence system was evaluated according to the “Expression of Uncertainty in Measurement” [31].

### 4.1. Uncertainty Analysis of Diffusivity in the Volumetric Analysis Method

Table 1 shows the uncertainty factors and expanded uncertainties for the measurements of hydrogen diffusion. The uncertainties in the diffusivity measurements were primarily due to the inconsistency of repeated measurements, changes in the sample volumes, and the standard deviations between the data and Equation (4). The type A uncertainty resulting from the repeated diffusivity measurements was determined with three measurements. All type B uncertainty contributions, except for the contribution from the resolution of the graduated cylinder, were obtained by dividing the uncertainty by a factor of $\sqrt{3}$ under the assumption of a rectangular distribution. The uncertainty due to mass measurements of the samples was estimated based on the accuracy of the electronic balance employed. After charging the sample, the maximum change in the dimensions of the sample reached 2.5%. Thus, the type B uncertainty resulting from uneven sample volumes was acquired. The standard deviation between the data for the amount of hydrogen versus time and the fit obtained with Equation (4) ranged from 0.5–3.0%, depending on the samples. By considering that the maximum deviation was 3%, type B uncertainty was obtained. The accuracy of the graduated cylinder was 0.5%; thus, the type B uncertainty was 0.3%. When a 10 mL graduated cylinder was used, the minimum readable scale was 0.1 mL, which corresponded to a 1% relative uncertainty. The resolution was half of this minimum value. Therefore, the type B uncertainty caused by the resolution was determined to be 0.2% by dividing by $\sqrt{6}$ for a triangular probability distribution. The accuracy of the manometer used for the exposed pressure measurements was 1%, which corresponded to Grade A. Thus, type B uncertainty was obtained. The variations in temperature and pressure during the measurements performed in the laboratory amounted to ±0.2 °C and ±5 hPa, respectively. However, the type B uncertainty arising from variations in temperature and atmospheric pressure was minimized to 0.2% by programmable compensation.

The combined standard uncertainty was expressed as a root sum of squares for the uncertainty factors because they were independent. The relative expanded uncertainty was obtained by assuming a normal distribution and multiplying the combined standard uncertainty by a coverage factor of 2.1 for the 95% confidence level. The estimated expanded uncertainty for the diffusivity was below 8.8%. In addition, the expanded uncertainty of the activation energy for diffusion in the VAM was estimated to be 10% by adding the uncertainty factor of 2.5%, which originated from the slope deviation in Figure 5, to the uncertainty factors, as shown in Table 1.

### 4.2. Uncertainty Analysis of Permeability in the Differential Pressure Method

Table 2 shows the uncertainty factors and expanded uncertainties for H_2_ permeability with the DPM. The dominant uncertainties in the permeability measurements were caused by the repeated measurements, maximum changes in the sample thickness, and variation in the permeation area contacting H_2_. The type A uncertainty for the repeated permeability measurements was obtained with five measurements. All type B uncertainty contributions were obtained by dividing the uncertainty by $\sqrt{3}$ for a rectangular distribution. The uncertainty due to the temperature measurements was estimated from the resolution and accuracy of the thermocouple used (k-type). In the experimental temperature range of −20 °C to 60 °C, the minimum reading digit of the thermocouple was 0.1 °C. The resolution was half of this minimum value. Therefore, the type B uncertainty caused by the resolution was determined to be 0.01% after dividing by $\sqrt{3}$ for a rectangular distribution. The uncertainty for the accuracy of the thermocouple was equivalent to 0.03% over the experimental temperature range. In Figure 3b, two vacuum gauges were installed on the upper and lower cell parts. The accuracy was 0.2%, according to the vacuum gauge certificate. Therefore, the type B uncertainty was 0.1% for the vacuum gauge. The uncertainty for the measured volume of the permeated side in the cell was determined to be 0.01% from the standard deviation between the data and *dp*/*dt* with Equation (5) with leak standard equipment. The uncertainty of the thickness measurements for the sample was 0.8%, depending on the calibration certificate, accuracy, and resolution of the Vernier caliper. After mounting the sample between the sample holders, the maximum change in the sample thickness reached 2%. Thus, the type B uncertainty for the change in the sample thickness was 1.2% based on the rectangular distribution. When manufacturing the permeation cell, the permeation area was designed to be 800 mm^2^. However, the O-ring seal changed the area by at most 40 mm^2^. Thus, the variation in the permeation area amounted to a maximum value of 5%. The type B uncertainty for the variation in the permeated area was 2.9% based on the rectangular distribution.

The combined standard uncertainty was expressed as a root sum of squares for the uncertainty factors because they were independent of each other. The relative expanded uncertainty was obtained by assuming a normal distribution and multiplying the combined standard uncertainty by a coverage factor of 2.6 for the 95% confidence level. The estimated expanded uncertainty for the H_2_ permeability was nearly 11%. In addition, the expanded uncertainty of the activation energy for permeation in the DPM was estimated to be 13% by including the uncertainty factor of 2.5% that originated from the deviation of the slope in the permeation versus reciprocal temperature plot, as shown in Table 2.

## 5. Conclusions

We established two different systems for measuring the temperature dependence of the H_2_ diffusivity and the permeabilities of NBR polymer composites by utilizing a volumetric analysis method with a modified diffusion analysis program and a differential pressure method with a permeation cell, respectively. The uncertainties for the two methods were analyzed by quantifying the possible uncertainties resulting from the overall measurements with the two systems. According to the temperature dependence of diffusion in the volumetric analysis method and the temperature dependence of permeation in the differential pressure method, the activation energies for the two methods were determined as a function of filler content and species. The activation energies for diffusion from the NBR polymer composites investigated in the volumetric analysis method were consistent with those for permeation determined by the differential pressure method. This finding implied that the H_2_ solubility was unchanged over a temperature range of 258 K to 323 K. The activation energies for diffusion in the NBR polymer composites blended with HAF CB and silica fillers increased with increasing filler content; those for the NBR polymer composites blended with MT CB fillers decreased with increasing filler content. The activation energies for the NBR composites filled with MT CB and silica were similar to the activation energy for neat NBR within the uncertainty level. This activation energy was very similar to the filler-dependent glass transition temperature determined by dynamic mechanical analysis.

In conclusion, the established temperature-dependent systems were utilized for investigations of dynamic diffusion/permeation in polymer specimens. From the activation energy and pre-exponential factors, we could predict the permeation parameters (diffusivity and permeation) at temperatures above the glass transition.

## Figures and Tables

**Figure 1 polymers-16-00280-f001:**
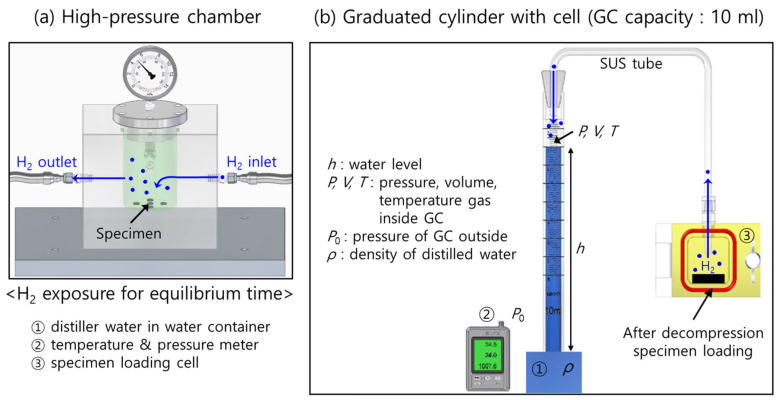
Volumetric measurements for hydrogen employing a graduated cylinder with a cell after high-pressure exposure and decompression: (**a**) specimen exposed in a high-pressure chamber. The light gray cubic-shaped box indicates the supporting body containing the cylindrical-shaped high-pressure chamber, manufactured from SUS 316 material to withstand pressures up to 20 MPa. The dark gray box below the chamber is the shelf plate on which the chamber is horizontally placed during the hydrogen exposure and (**b**) after decompression in the chamber and specimen loading into the cell. The H_2_ emission measurement was conducted with a graduated cylinder partially immersed in a water container. The blue color in the cylinder indicates distilled water.

**Figure 2 polymers-16-00280-f002:**
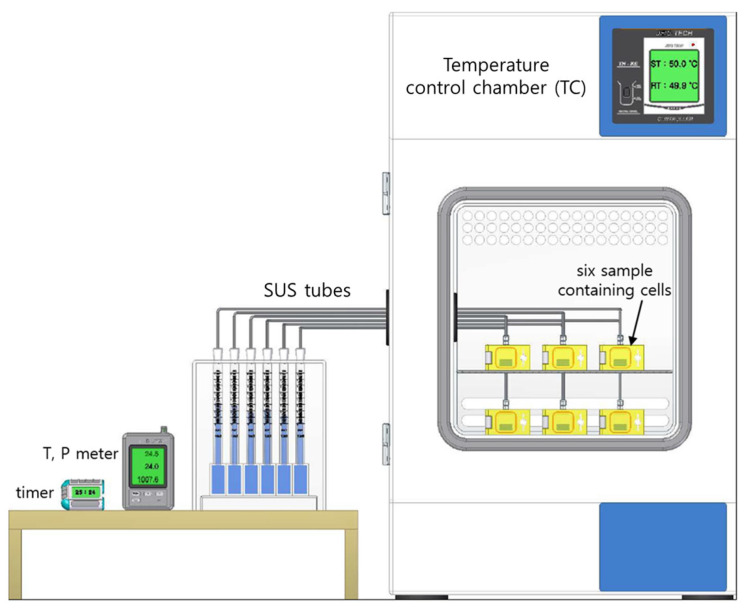
Temperature-controlled chamber for six parallel volumetric measurements with six graduated cylinders after high-pressure exposure and decompression. There are six sample-containing cells in the temperature control chamber. The blue in the cylinder indicates distilled water.

**Figure 3 polymers-16-00280-f003:**
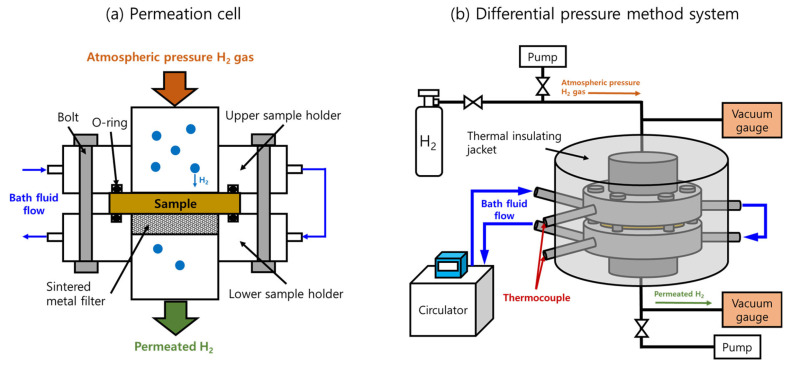
Schematic of the temperature-control system of the DPM: (**a**) front view of the internal cross-section for the permeation cell and (**b**) overall temperature-controlled system with the bath fluid circulator outside the permeation cell.

**Figure 4 polymers-16-00280-f004:**
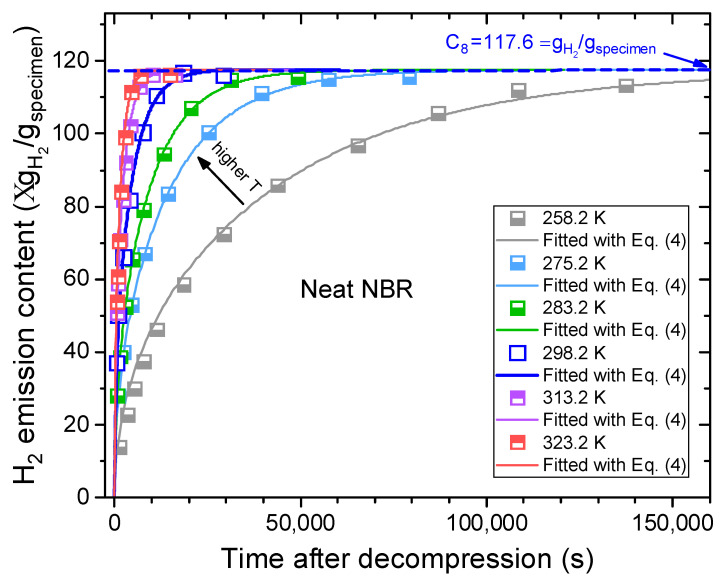
H_2_ emission content versus time for neat NBR at different temperatures after hydrogen exposure at 7 MPa. The corresponding solid lines are the least-squares fits to Equation (4) with the diffusion analysis program. The blue arrow on the dashed blue line indicates the hydrogen uptake at infinite time.

**Figure 5 polymers-16-00280-f005:**
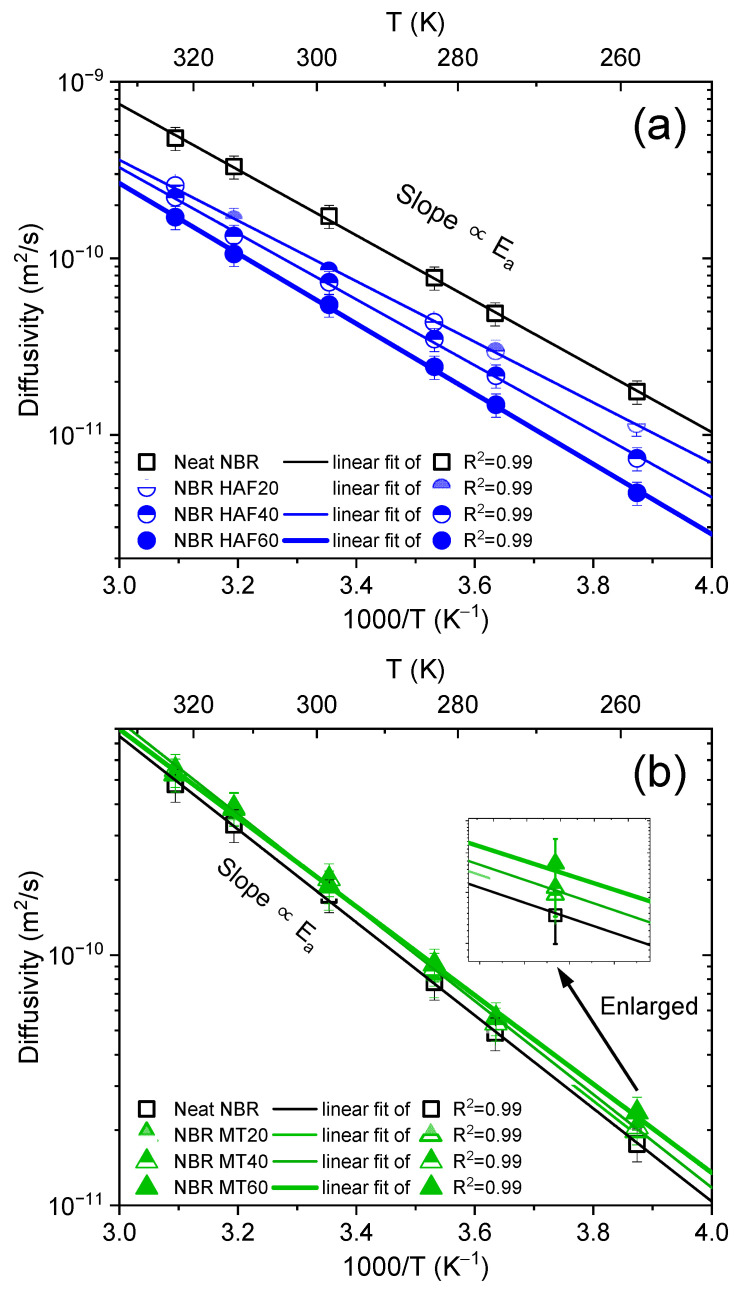
Hydrogen diffusivity versus reciprocal temperature for neat NBR and NBR composites blended with fillers: (**a**) HAF CB filler, (**b**) MT CB filler, and (**c**) silica filler. The slopes of the linear fits indicated the activation energies for diffusion. The diffusivity result for neat NBR is included in all panels for comparison with those of the NBR composites with fillers. The diffusivities for the NBR MT series and the NBR silica series were almost the same as for neat NBR within the measurement uncertainty, as shown in the enlarged graph. The error bars represent the expanded uncertainty (8.8%) in the diffusivity.

**Figure 8 polymers-16-00280-f008:**
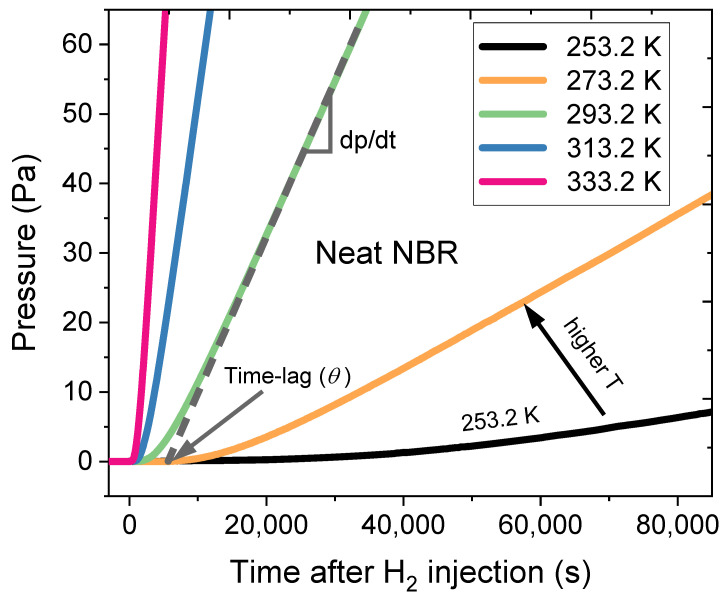
Hydrogen pressure measured on the permeated side versus time after H_2_ injection for neat NBR at different temperatures. The slope in the permeation curves was obtained after hydrogen injection started (t = 0).

**Figure 9 polymers-16-00280-f009:**
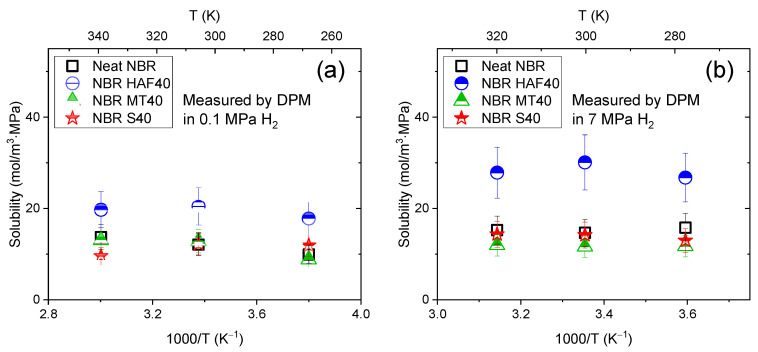
Temperature-independent solubility in the DPM measurements for the neat NBR and NBR composites filled with HAF CB, MT CB, and silica at 40 phr at three different temperatures: (**a**) measured at 0.1 MPa H_2_ and (**b**) measured at 7 MPa H_2_.

**Figure 10 polymers-16-00280-f010:**
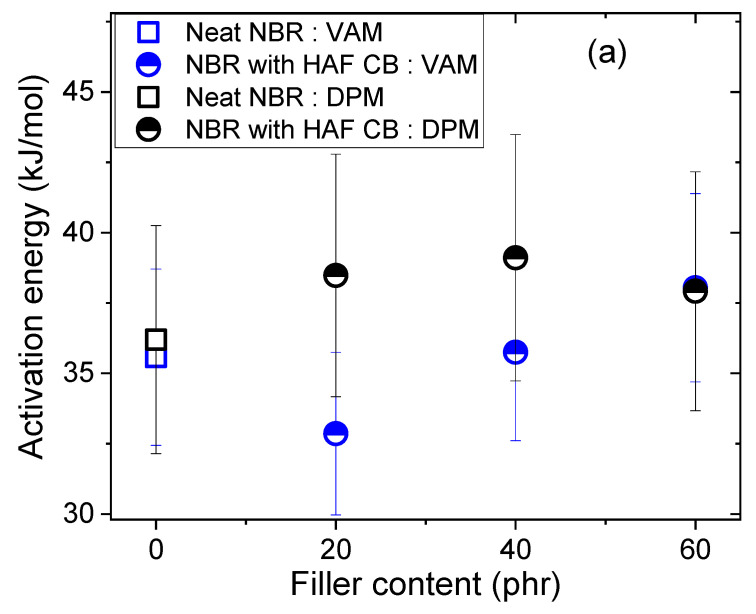
Comparison of the activation energies determined in the VAM with those obtained in the DPM: (**a**) neat NBR and NBR composites filled with HAF CB; (**b**) neat NBR and NBR composites filled with MT CB; and (**c**) neat NBR and NBR composites filled with silica. The results for neat NBR are contained in all panels for comparison with those of the NBR composites blended with fillers. The error bars representing the expanded uncertainty for the activation energy in DPM are 13%, as estimated in the uncertainty analysis.

**Table 1 polymers-16-00280-t001:** Uncertainty sources and expanded uncertainties for the temperature dependence system used to measure the H_2_ diffusivity in the VAM.

Uncertainty Factor	Relative Value (%)
Repeated measurements	3.5
Accuracy of the electronic balance	0.1
Change in the sample volume	1.4
Standard deviation between the data and Equation (4)	1.7
Accuracy of the graduated cylinder	0.3
Resolution of the graduated cylinder	0.2
Accuracy of the manometer	0.6
Variations in the temperature/pressure	0.2
Combined standard uncertainty, *u*_c_	4.2
Coverage factor, *k*	2.1
Expanded uncertainty, *U* = *ku*_c_	8.8

**Table 2 polymers-16-00280-t002:** Uncertainty sources and expanded uncertainties for the temperature dependence system used in measuring the H_2_ permeability in the DPM.

Uncertainty Factor	Relative Value (%)
Repeated measurements	2.9
Resolution of the thermocouple	0.01
Accuracy of the thermocouple	0.03
Accuracy of the vacuum gauge	0.1
Volume of the permeated side in the cell	0.1
Standard deviation between the data and the *dp*/*dt* in Equation (5)	0.01
Thickness measurement for the sample	0.8
Maximum change in the sample thickness	1.2
Variation in the permeation area contacting H2	2.9
Combined standard uncertainty, *u*_c_	4.3
Coverage factor, *k*	2.6
Expanded uncertainty, *U* = *ku*_c_	11.2

## Data Availability

The data used to support the findings of this study are available from the corresponding author upon request.

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
