# Peer review of "Comparison of Two Methods for Measuring the Temperature Dependence of H2 Permeation Parameters in Nitrile Butadiene Rubber Polymer Composites Blended with Fillers: The Volumetric Analysis Method and the Differential Pressure Method"

_polymers, 2024, doi:10.3390/polym16020280_

Round 1

Reviewer 1 Report

Comments and Suggestions for Authors

This is a nicely written manuscript, establishing the temperature dependence of H2 diffusivity in polymer composites. There are a few points that the authors need to address before it gets accepted.

1. In the Introduction section, the authors need to provide a brief review on the reported experimental methods for measuring H2 diffusivity/permeation in polymer composites. This would be helpful for the readers to appreciate the motivation of this work.

2. The authors need to provide more details about the experiments, for examples, sample preparation methods (compounding), and glass transition temperature measurements (DMA).

Reviewer 2 Report

Comments and Suggestions for Authors

Overall, this reviewer finds this to be a clearly written paper.  Some experimental details are lacking for Tg measurement (see comment #5 below). 

Comments for revision:

1. p2 - when expressing the range of temperatures relevant for H2 transport, it would be clear to express the range as "-50C to 90C" (instead of "-50C-90C")

2. Figs 1 and 2 are very good and highly detailed, but the small details are only readable if the figures are very high resolution or very large in the document.  In the copy for review, the resolution was not high enough to see the detail.  The figures could be revised to be more schematic, or could be re-uploaded at a higher resolution. 

3. The use of single colours in Fig 5 (a-c) makes it difficult to distinguish between the filler contents.  Here the authors could consider using different symbols to show different filler content.  

4. Did the authors consider possible reasons for why the diffusivity of S-filled NBR appears to increase with filler content? 

5. The paper lacks an adequate description of how Tg was measured (results presented in Fig 7)

6. In Fig 8, the curves are shown outside the graph area.

7. Fig 10a - error bars for 20phr specimen covered by legend box.

8. In the VAM, is there any consideration for the solubility of H2 in water, or is this assumed to be negligible?

Round 2

Reviewer 2 Report

Comments and Suggestions for Authors

I have no further comments, and feel that this paper is ready for publication.